

# Seasonal variability of intermediate water masses in the Gulf of Cadiz: implications of the Antarctic and Subarctic seesaw model

Roque, David [1], Parras-Berrocal, Ivan[2], Bruno, Miguel [2], Sánchez-Leal, Ricardo [3], Hernández-Molina, Francisco Javier [4]

[1] Institute of Marine Sciences of Andalusia (ICMAN-CSIC), University Campus Rio San Pedro. Puerto Real, Cádiz11510, Spain.
[2] Applied Physics Department, University of Cádiz, Puerto Real, Cádiz 11130, Spain.
[3] Spanish Institute of Oceanography (IEO), Cádiz Oceanographic Centre, Muelle de Levante, Puerto Pesquero, Cádiz 11006, Spain.

[4] Department of Earth Sciences, Royal Holloway, University of London, Egham, Surrey TW20 0EX, UK.

*Correspondence to: David Roque (david.roque@icman.csic.es)*

**Abstract.** Global circulation of intermediate water masses has been extensively studied; however, its regional and local circulation along continental margins and variability and implications on sea floor morphologies are still not well known. In this study the intermediate water mass variability in the Gulf of Cádiz and adjacent areas has been analysed and its implications discussed. Remarkable seasonal variations of the Antarctic Intermediate Water (AAIW) and the Subarctic Intermediate Water (SAIW) are determined. During autumn a greater presence of the AAIW seems to be related to a reduction in the presence of SAIW and Eastern North Atlantic Central Water (ENACW). This interaction also affects the Mediterranean Outflow Water (MOW), which is pushed by the AAIW toward the upper continental slope. In the rest of the seasons, the SAIW is the predominant water mass reducing the presence of the AAIW. This seasonal variability for the predominance of these intermediate water masses is explained by a novel model based on the concatenation of several wind-driven processes acting during the different seasons. Our finding is important for a better understanding of regional intermediate water mass variability with implications in the Atlantic Meridional Overturning Circulation (AMOC) but further research is needed in order to decode their changes during the geological past and their role, especially related to the AAIW, in controlling both the morphology and the sedimentation along the continental slopes.

Key words: Intermediate Water Mass circulation; Seasonal variability; Antarctic Intermediate Water; Subarctic Intermediate Water; Mediterranean Outflow Water; Gulf of Cádiz; Atlantic meridional overturning circulation.





## 1 Introduction

The intermediate water masses, their circulation and associated oceanographic processes are essential to understanding meridional heat, freshwater and nutrient transports, the Thermohaline Circulation (THC) and the Meridional Overturning Circulation (MOC) and oceanic sink for anthropogenic CO2 (Sabine et al. 2004). Although their circulation at global scale

has been extensively studied (Rahmstorf 2006), their spreading, interactions and the regional implications along continental margins are still not well known. At a global scale, the Antarctic Intermediate Water (AAIW) is one of the most relevant intermediate waters, since it is an important component of the upper branch of the MOC in the Southern Hemisphere subtropical gyre (Pahnke, Goldstein, and Hemming 2008; Talley 2003). It is characterised by a salinity minimum and high nutrient concentration between 500-800 m to ca. 1500 m depth, with its main core located at 900 m (Fig. 1). The AAIW

spreads along the eastern boundaries of all major oceans, although its original characteristics are best observed in the Southern Hemisphere (Stramma 1999).

Several authors have reported a significant time variability of AAIW presence in the North Atlantic, enhanced during cold periods and weakens and shoals during warmer periods (Oppo 2000; Curry and Oppo 2005; Jung et al. 2010; Makou, Oppo, and Curry 2010; Muratli et al. 2010; Wainer et al. 2012). The AAIW is well identified year-round in the eastern boundary of

the North Atlantic. It spreads along the African coastline up to approximately 28.5 °N but during autumn it can extend up to 32 °N (Tsuchiya, Talley, and McCartney 1992; Machín and Pelegrí 2009) and even into the Gulf of Cádiz (GoC)(Cabeçadas, José Brogueira, and Gonçalves 2002; Louarn and Morin 2011; Hernández-Molina et al. 2014).

Traditionally, only two intermediate water masses are considered in the Gulf of Cádiz (Ambar and Howe, 1979; Baringer and Price, 1997; H M van Aken, 2000): the Eastern North Atlantic Central Water (ENACW) and the Mediterranean Outflow

Water (MOW) (Fig. 1). The ENACW is formed by winter cooling of saline surface water near the European ocean margin west and northwest of Spain and Portugal (Pollard and Pu 1985). From here, it circulates southward penetrating into the Gulf of Cádiz in the 250 - 700 m depth range. The ENACW interacts actively with the MOW close to the western Strait of Gibraltar (Louarn and Morin 2011) and with the upper MOW core further west, throughout the Gulf of Cádiz continental slope (Louarn and Morin 2011; M.J. Bellanco and R.F. Sánchez-Leal 2016).

The AAIW spreads northeastwards following the Moroccan and Mauritanian continental slopes. A westward branch enters the Gulf of Cádiz, where it interacts with the MOW, perhaps restricting the spreading of the latter, especially between summer and winter (Cabeçadas et al., 2002; Knoll et al., 2002; Louarn & Morin, 2011). This interaction causes the increment in AAIW nutrient and dissolved oxygen concentrations due to mixing with the MOW, particularly between 35° N and 37° N ( M. van Aken 2000).

Another important intermediate water mass in the Gulf of Cádiz is the Subarctic Intermediate Water (SAIW). It is formed by winter cooling of surface water in three main regions: a) North Western North Atlantic, north of the subarctic front, b) the Porcupine Seabight, and c) northern Bay of Biscay (Van Aken 2001). A significant part of the variety of water formed in the latter two regions, progresses below the North Atlantic Current and flows towards the Eastern Atlantic margin, penetrating



into the Gulf of Cádiz from a south-southwestern direction and moving in the depth range of 400-800 m (Cabeçadas et al. 2002).

The SAIW and the ENACW have been traditionally referred to as the lower and upper ENACW ( Louarn and Morin 2011; Carracedo et al. 2014). Following H M van Aken 2000; Van Aken 2001, we use the former denominations. Currently, there

is a lack of information about the seasonal variability of these intermediate water masses in the region. M.J. Bellanco and R.F. Sánchez-Leal 2016 analysed the interaction between the upper MOW core and the ENACW on the upper Gulf of Cádiz continental slope. The authors suggested an enhanced upper MOW core and a less stratified ENACW in winter, resulting in an upslope expansion of the MOW core and a more effective mixing with the overlaying ENACW. In contrast, they noted a weakened and deeper MOW core and a thicker ENACW during late summer and early autumn.

The study by Machín and Pelegrí 2009 is relevant for consideration of the seasonal variations of the AAIW water within the Gulf of Cádiz. Although focussed on the Canary Islands region, the authors reported a clear AAIW increase during autumn when compared with other seasons. The question whether this AAIW increase has implications on the other water masses circulating in the region is addressed in the present paper. Therefore, our main aims are: a) to determine the AAIW extension and variability near the Gulf of Cádiz; b) to identify its interplays with other intermediate water masses; c) to hypothesise

about its implication on the continental margin morphology. We analyse a large conductivity, temperature, and depth (CTD) and oxygen concentration profile database extracted from various providers, complemented with the World Ocean Atlas (WOA13).

## 2. Data set and methodology

### 2.1 The data set

In order to identify water masses coming from remote locations we considered a study area spanning from 30°-40° North and 15°-5.5° West (Fig. 2). The domain includes the Gulf of Cádiz and surrounding areas into the Gulf of Cádiz. We have gathered all available observations from WODB, WOCE, MEDATLAS, CORIOLIS, SEADATANET databases, complemented with data from other individual cruises, mostly carried out by the University of Cádiz (Golfo 2001, Medout, Estrecho 2008). The temporal distribution of profiles span from 1900-2013, although the bulk of the data was acquired after

1950. The resulting product consists of a database including vertical profiles of temperature, salinity, density and dissolved oxygen concentration. The latter will be used to discern the AAIW from the SAIW and the ENACW ( Machín and Pelegrí 2009; Louarn and Morin 2011; Carracedo et al. 2014; Hernández-Molina et al. 2014). We also included the World Oceanographic Atlas monthly objectively analysed (1° grid) climatological fields (https://www.nodc.noaa.gov/OC5/woa13/). To study the seasonal variability, we grouped observations into 3-month seasonal bins, starting in December (i.e., winter

spans December through February, spring spans March through April, and so on). Since the analysis will be focused on intermediate water masses, we only used the observations taken within the 400-1000 m depth range. Figures 2 and 3 show



the seasonal distribution of vertical profiles as well as the WOA data nodes. The figures reveal the homogeneous spatial distribution of profiles.

| Data base | Temporal range | Link |
|---|---|---|
| WOD | 1900-2013 | https://www.nodc.noaa.gov/OC5/WOD13/data13geo.html |
| WOCE | 1990-1999 | http://www.ewoce.org/data/index.html |
| Medatlas | 1900-2010 | https://odv.awi.de/data/ocean/medatlasii/ |
| Coriolis | 1990-2012 | https://odv.awi.de/data/ocean/coriolis-cora-3-dataset/ |
| Seadatanet | 1930-2010 | https://www.seadatanet.org/ |
| WOA | 1955-2009 | https://www.nodc.noaa.gov/OC5/SELECT/woaselect/woaselect.html |

Table 1. Details of the different databases used in the analysis.

In addition, to examine the possible mechanisms behind the time variability of water masses we used monthly wind data taken from the ERA-Interim reanalysis (Dee et al. 2011) spanning 1979-2018, downloaded from https://www.ecmwf.int/en/forecasts/datasets/archive-datasets/browse-reanalysis-datasets. The spatial data resolution is 0.125 x 0.125 degrees.

**2.2 Optimum Multiparameter Analysis**

In order to quantify the percentage in which each water mass was presented in the studied area we applied an Optimum Multiparameter Analysis (OMP) to each dataset. This technique assumes that the value of any sea water property, measured at each depth in the ocean, is the result of the mixing of a certain number of water masses present in the region (Tomczak and Large 1989). It has become a standard tool in oceanography to resolve water mass mixing in regional scales studies ( Poole and Tomczak 1999; Álvarez et al. 2004; Johnson 2008; Louarn and Morin 2011; Pardo et al. 2012).

Mathematically, it implies solving an overdetermined system of linear equation which is composed of (n+1) equations, 'n' being the number of unknown variables, the fraction (percentage) of each water mass present in the mixing. As an example, for a problem with four water masses the following system of equations would apply:

$$x_1T_1+x_2T_2+x_3T_3+x_4T_4 = T_o \qquad (1)$$
$$x_1S_1+x_2S_2+x_3S_3+x_4S_4 = S_o \qquad (2)$$
$$x_1C_1+x_2C_2+x_3C_3+x_4C_4 = C_o \qquad (3)$$
$$x_1+x_2+x_3+x_4 = 1 \qquad (4)$$





where the $x_{is}$ are the fraction (percentage) of each water mass present (i varying from 1 to 4); $T_o$, $S_o$ and $C_o$, the observed temperature, salinity and concentration of some conservative property (i.e. oxygen) at each point; $T_{is}$, $S_{is}$ and $C_{is}$ are the characteristic values of properties for each water mass type.

The analysis is restricted to the depth range of 400-1000 m and considers the following four water mass types: AAIW, SAIW, ENACW and MOW. The characteristic values of properties for these water masses in the Gulf of Cádiz region have been taken from (Louarn and Morin 2011) (Table 2). Note that the difference in S and T between the AAIW and SAIW is very small and can only be distinguished by their oxygen concentration. For this reason we have chosen oxygen as the other conservative property in the OMP analysis.

| WATER MASS TYPE | Temperature (° C) | Salinity | Oxygen (µmol/Kg) |
| --- | --- | --- | --- |
| AAIW | 10.25 | 35.62 | 167 |
| ENACW | 16.374 | 36.289 | 225.93 |
| SAIW | 10.659 | 35.434 | 185.2 |
| MOW | 13.9 | 36.64 | 180 |

Table 2. Values of physical-chemical variables taken for the different water masses.

## 2.3 Principal component analysis

The principal component analysis (PCA) is a statistical procedure that allows the transformation of a set of observations, which contain variables possibly correlated among them, into a set of values of uncorrelated variables called principal components (Pearson 1901). When the problem under study contains a large number of observations the maximum number of principal components to be sought for is equal to the number of variables. This transformation is defined in such a way that the first principal component has the largest possible variance, and the successive components progressively reduce their variance.

This technique has been applied on the fractions of the four water mass types determined in the OMP analysis previously described, in order to identify common patterns of behaviour among them while they change with the seasons. Therefore, the observed variables of our PCA problem are these fractions of each water mass type: $x_1$, $x_2$, $x_3$ and $x_4$. Subsequently, the values of these variables at each spatial location and in each season are arranged into four columns and the resulting file is subjected to the PCA analysis. The resulting principal components (PCs) may be computed as:

$$PC_1 = C_1^1 x_1 + C_1^2 x_2 + C_1^3 x_3 + C_1^4 x_4 \qquad (5)$$

$$PC_2 = C_2^1 x_1 + C_2^2 x_2 + C_2^3 x_3 + C_2^4 x_4 \qquad (6)$$

$$PC_3 = C_3^1 x_1 + C_3^2 x_2 + C_3^3 x_3 + C_3^4 x_4 \qquad (7)$$

$$PC_4 = C_4^1 x_1 + C_4^2 x_2 + C_4^3 x_3 + C_4^4 x_4 \qquad (8)$$



That is, as a linear combination of the observed variables $x_{is}$ weighted by a set of coefficients $C_j^i$, in which, 'j' and 'i' stands for, respectively, a given PC and a given variable. Note that each PC offers a value for each observation and these PC values are assigned to the same location and time as each observation.

At the same time, matrix algebra allows us to write the inverse problem:

$$x_1 = C_1^1 \, PC_1 + C_2^1 \, PC_2 + C_3^1 \, PC_3 + C_4^1 \, PC_4 \tag{9}$$

$$x_2 = C_1^2 \, PC_1 + C_2^2 \, PC_2 + C_3^2 \, PC_3 + C_4^2 \, PC_4 \tag{10}$$

$$x_3 = C_1^3 \, PC_1 + C_2^3 \, PC_2 + C_3^3 \, PC_3 + C_4^3 \, PC_4 \tag{11}$$

$$x_4 = C_1^4 \, PC_1 + C_2^4 \, PC_2 + C_3^4 \, PC_3 + C_4^4 \, PC_4 \tag{12}$$

where each of the observed values of the variables $x_i$ may be expressed as a linear combination of the resolved PCs weighted by the coefficients $C_j^i$ already introduced.

Now the absolute values that take these coefficients for a given PC along the different variables enable us to identify which of them are related to that PC. In order to visualise the temporal patterns contained in the resolved PCAs, it is worthwhile to separate the products ($C_j^i \cdot PC_j$) corresponding to each climatic season for each $PC_j$. Later, we will be able to map the seasonal variability contained in each PC and assess how intense is its manifestation in the different spatial locations of the studied area.

## 3. Results

### 3.1 Spatial distribution of physical- chemical properties

As a first step of the data analysis, several spatial representations of the different variables have been elaborated:

a. Representation of the spatial distribution of the variables on the isopycnic surface of 27.5 which best characterise the position of the AAIW core in the region (Van Aken 2001; Cabeçadas et al. 2002; Louarn and Morin 2011; Hernández-Molina et al. 2014).

b. Representation of vertical sections of the variables along the meridional section at a latitude of 36º N which meets two important conditions: (i) To allow an adequate identification of the AAIW pathway by the Gulf of Cádiz and (ii) To contain a high number of profiles covering as homogeneously as possible, during the different seasons, the given vertical sections.

### 3.1.1 Spatial distribution on the 27.5 isopycnic surface

Figures 2 and 3 show, respectively, the distributions of salinity and oxygen concentration on the 27.5 isopycnic surface, for the seasonal values contained in the observed data (upper row) and for the monthly values provided by the WOA database (following three rows).

Concerning the observed data, we found low oxygen concentration values (Fig. 3) coincident with low salinity values (Fig. 2), attributable to the AAIW presence (see Table 2), penetrating from lower latitudes into the Gulf of Cádiz in all seasons,



although it is remarkable that this penetration is more accentuated and closer to the African coast in autumn. Note also that the lowest presence of AAIW is found in spring (Fig. 2 and 3).

The monthly distributions shown by the WOA data on the 27.5 isopycnic, roughly agree with the previous commented behaviour. It must be noted that in this case these data do not offer such a detailed spatial resolution of the region closer to the continental margin as the observed data do. However, these maps allow us to gain insight into the inter-annual variability of the AAIW distribution close to the GoC. In this sense, we can see that the major entrance of the AAIW close to the African coast occurs in November. At the same time, we can see that the spreading of the AAIW toward the GoC is more restricted during the spring months. In addition, in May, June and July this spreading is re-stablished, albeit to a lesser extent than in autumn.

### 3.1.2 Vertical section distribution in zonal direction at latitude 36 º N

Figure 4 shows the zonal distribution of dissolved oxygen, salinity and temperature along 36º N. Low oxygen values characteristic of the AAIW occur year-long in the 700-1200 m depth range. As suggested before, the lowest oxygen concentrations occur in autumn close to the African continental slope. This increased AAIW presence close to the African Coast in autumn was already reported by ( Tsuchiya et al. 1992; Knoll et al. 2002; Machín and Pelegrí 2009, 2016), although at lower latitudes.

In summer, the low oxygen zone spreads zonally, extending to Cape San Vicente (Fig. 4). In winter, the low oxygen signal is less evident and relative oxygen values higher, perhaps suggesting weaker AAIW presence in winter.

### 3.2 OMP analysis

The AAIW is present as an intermediate water mass extending along the middle slope from northern Morocco to the Gulf of Cádiz year-round. There seems to be a certain intra-annual variability, with enhanced AAIW presence in autumn. To elucidate the pattern, we applied the OMP analysis to the dataset of vertical profiles in the target depth range (400-1000 m). Subsequently, we binned the results into 9 1ºx1º boxes (Fig. 5). As suggested before, the highest AAIW percentages are found in autumn. Only box C (the closest to the continent) and G (the farthest from the coast) show AAIW percentages below 40 %. The SAIW shows its lower percentages (less than 10 %) in boxes closer to the coast (from E to I) during autumn while the higher values are found in winter. The ENACW in the closer to coast boxes shows its higher values in spring and summer while the MOW shows these in winter and autumn. These results show the relationship among the different water masses. On the one hand, in autumn the AAIW displaces the SAIW while the reverse situation occurs in winter. On the other hand, the alternation in predominance of these two water masses seems to be related to (i) a major confinement of the MOW toward the coast in autumn (by the enhanced AAIW presence) and winter (by the SAIW) and (ii) a displacement toward offshore of the ENACW in the same seasons.

Regarding the OMP analysis on the WOA data, we have selected data located around a zonal section located at a latitude of 36 º N (Fig. 5) in order to assure a clear identification of the AAIW coming from the Canary Islands latitude where this



water mass have been previously identified close to the African coast (Machín and Pelegrí, 2009; 2016; Louarn and Morin, 2011). Similar to the case of the observed profiles only data in the depth range of 400-1000 m have been considered. The water mass percentages have been computed for each month along the climatic year. Figure 5 shows the vertical sections of computed percentages for the AAIW for February, April, July and November. The stronger AAIW presence occurs in

November, when it is found near the African continent. We observe a weaker AAIW presence from February to April. The OMP computation assumes that the autumn increase of the AAIW goes along with a reduction of the SAIW fraction, hence illustrating a sort of competitive seesaw between the two water masses. Figure 9a shows the climatic monthly values of the averaged percentages over the vertical section percentages of the four water masses. It can be seen that the SAIW is the most predominant water throughout most of the year, namely greater than 70 % for all months, with the exception of November

when the AAIW rises to 40 %.

### 3.3 PC analysis

In order to carry out a better assessment of the seasonal variations of water mass percentages and the interrelation between the different water masses, we have applied a PCA on the set of computed percentages. As explained in section two, this technique will allow us to identify common patterns in the seasonal variations of these percentages in the nine boxes located

in the GoC shown in Figure 5.

In Figures 6 and 7, the results of this analysis are shown. The first component (Fig. 6) shows a clear interaction between the AAIW and SAIW. In autumn a greater presence of the AAIW, coming from lower latitudes, seems to be related to a reduction in the presence of the SAIW coming from higher latitudes. In addition, this first principal component also reflects a significant increase in the presence of the MOW in the box closest to the continental slope. The second principal

component (Fig. 7), picks up another clear interaction between the four water masses, but now it seems to be caused by a greater presence of the SAIW in winter that produces a displacement of the AAIW and NACW and again a confinement of the MOW against the continental slope.

The interpretation of results is sketched in Figure 8, which shows the likely preferential tracks of the different intermediate waters along the GoC in the different seasons. In autumn the AAIW penetrates closer to the continental slope while the

SAIW runs more displaced southward. Also, the ENACW and MOW are pushed toward the coast. In winter the situation is reversed regarding AAIW and SAIW behaviour. Now the SAIW is flowing closest to the continental slope, the AAIW is displaced southward and once again the ENACW and MOW are confined toward the continental slope. The situation in spring and summer is similar to the one in winter.

### 3.4 Wind forcing in the north Atlantic and intermediate water presence in the Gulf of Cádiz

Variability of the meridional transport in the North Atlantic is linked with two important wind driven mechanisms ( Machín and Pelegrí 2009; Barrier et al. 2014):

The meridional Sverdrup transport:





$$M_y = \frac{1}{\rho_0 \beta_0} \left[ \frac{\partial \tau_y}{\partial x} - \frac{\partial \tau_x}{\partial y} \right] \tag{13}$$

and the Ekman pumping:

$$w_E = \frac{1}{\rho_0} \left[ \frac{\partial}{\partial x} \left( \frac{\tau_y}{f} \right) - \frac{\partial}{\partial y} \left( \frac{\tau_x}{f} \right) \right] \tag{14}$$

Where $\tau_x$ and $\tau_y$ are, respectively, the zonal and meridional components of wind stress; ρ0 is sea water density; f=f0 + β0y is Coriolis parameter, where β0 is the variation with latitude of $f$ in the β plane.

We computed both magnitudes from wind velocity data provided by ERA-Interim reanalysis spanning the period 1979-2018. Figures 10 and 11 show the meridional Sverdrup transport and Ekman pumping for January and July, the two seasonal opposites in a climatological year. They are shown for the region of the North Atlantic spanning the latitudes 10º N to 70º N

and for the zoomed region focussed on the Gulf of Cádiz.

The meridional Sverdrup transport peaks close to the eastern coast (Figure 10). This suggests small-scale horizontal wind stress gradients near the continent. The Sverdrup transport along the eastern coast of Portugal is southward year-round, with highest (lowest) values in January (July). Along the African coast this transport is northward year-round, with highest (lowest) values in July (January). More detailed information about the seasonal variations of these transports can be found in

Figure 9b where the mean monthly climatological values of Sverdrup transport are shown at three points close to the coast, namely at latitudes 30º N, 35º N and 40º N. At 30º N meridional transport is positive (northward) all year showing an increase in summer with a peak in July. This northward transport close to the African coast may contribute to the northward progression of the AAIW from latitudes as low as 10º up to at least 30º N. In contrast, the transport at 40º N is southward year-round, peaking in winter. Therefore, this behaviour could explain a special progression of the SAIW close to the

Portuguese coast from high latitudes down to the GoC during this season.

Concerning the Ekman pumping, Figure 11 shows its spatial distribution in the same two months of the climatological year, January and July, in the same domains considered in Figure 9. Note that the zones with negative values of the Ekman pumping roughly depict the location of the gyres within the basin. In winter (i.e. January) the zone where convergence is more important (negative Ekman pumping) within the subtropical gyre reaches latitudes as high as 50º while in summer (i.e.

July) this zone is more reduced and displaced to latitudes lesser than 30º. Therefore, in winter, a more developed subtropical gyre may favour the increasing transport of high latitude intermediate waters toward the Gulf of Cádiz while, in summer, a displacement toward the south gyre is not able to transport these high latitude intermediate waters toward the Gulf of Cádiz.

Besides the small scale variations in Ekman pumping, we observe generalised divergence near the eastern ocean margins all year-round. As the Ekman pumping is implicitly accounted for in the Sverdrup transport, we expect that divergence near the

continents may be held responsible for the local intensification of the meridional Sverdrup transport.

In order to present more detailed information about the time variability of the Ekman pumping at the latitude of the Gulf of Cádiz, Figure 9c shows the monthly climatological averages of this variable, inside the GoC, in a box limited by latitudes 35º and 36º N and longitudes 7º and 8º W, and outside the gulf, in a box limited by latitudes 35º and 36º N and longitudes



19º and 20º W. Inside the GoC there is permanent divergence (positive pumping) in the Ekman layer caused by a small-scale cyclonic wind pattern existing in this area. This divergence, which is more intense during summer, would favour the displacement of the water masses underneath the Ekman layer toward the gulf, exerting a suction effect on them. So, intermediate waters that arrive at a latitude close to the gulf could continue progressing northward thanks to this effect.

Outside the GoC there is a permanent convergence (negative pumping), which is more intense during winter and summer. It is expected that during these seasons the dominant downwelling existing in the Ekman layer of this zone would not favour the transport of intermediate waters in that direction. However, in spring and autumn the intensity of these convergences decreased significantly, which could diminish the capacity of blocking the transport of intermediate waters in that direction. In the following section we discuss all these preliminary results as well as the results attached in the related references in

order to depict a conceptual model that allows explanation of the mechanisms that control the fluctuation of intermediate water masses in the GoC.

## 4. Discussion

### 4.1 Extension and variability of the AAIW and its interrelation with other intermediate water masses

Presence of the AAIW in the GoC has been reported before (H. M. van Aken 2000; Cabeçadas et al. 2002; Brogueira,

Cabeçadas, and Gonçalves 2004; Louarn and Morin 2011; Preu et al. 2013; Hernández-Molina et al. 2014). There is also evidence of greater autumn AAIW presence near the African margin at least up to 32 °N ( Tsuchiya et al. 1992; Knoll et al. 2002; Machín and Pelegrí 2009). Until now, we remain uncertain whether this seasonal variability extends into the GoC. In section 3 we confirmed AAIW presence year-round along the middle slope from the northern Moroccan margin and into the GoC. We inferred certain seasonal AAIW variability featuring an autumnal approach to the continent, summer horizontal

spread and offshore separation. Winter and particularly spring are characterized by a weaker AAIW presence. Moreover, the results of the PCA on the water mass percentages allow the identification of clear interactions between the four water masses. In autumn a greater presence of the AAIW, coming from lower latitudes, seems to be related to a reduction in the presence of the SAIW and ENACW. This interaction also affects the MOW, which is pushed by the AAIW toward the continental slope. In winter, the SAIW is the predominant water mass reducing the presence of the AAIW and

ENACW and once again pushing the MOW toward the continental slope. The next step now is to investigate the physical mechanism that triggers these seasonal variations in the water mass presence. Machín and Pelegrí 2009 explained the arrival of the AAIW to latitudes around the Canary Islands as the divergences produced by the Ekman pumping close to the African coast, and the subsequent stretching of the water column below, promoting a northward movement of the intermediate waters in order to conserve its potential vorticity. However, the

identified divergence regions were located too far from the African coast to explain the observed confinement toward the coast of the observed AAIW core. Perhaps the spatial resolution of 1o of the wind data used was too coarse to capture the small scale variation of winds close to the continent. The importance of these small scale variation of winds has been





recognised in several studies (Kanzow et al. 2010; Machín et al. 2010; Pérez-Hernández et al. 2013; Velez-Belchi and Hernandez-Guerra 2017) in connection with the generation of baroclinic Rossby waves that propagate toward the interior ocean and may affect the Atlantic meridional overturning circulation (AMOC).

Barrier et al. (2014), who have analysed the response of the North Atlantic subtropical gyre to the North Atlantic Oscillation
(NAO) and other atmospheric regimes (Fig. 12), conclude that this response involved important changes in meridional transport of water masses in this basin. For instance, a positive/negative NAO displaces the subtropical gyre to a higher/lower latitude. A gyre centred at a higher latitude favours a southward meridional transport of high latitude water masses by the eastern boundary while a gyre centred at lower latitude does not favour this southward transport of water masses from higher latitudes (Fig. 12). In terms of the intermediate water masses we are dealing with, a positive NAO would
prevent the entrance of the AAIW toward the GoC in favour of a greater SAIW arrival here, while a negative NAO would weaken the southward transport of the latter and favour the spread of the AAIW toward the Gulf of Cádiz.

Keeping these previous results in mind and referring to the results presented in section 3.3, we will propose a mechanism based on the seasonal variation of wind forcing in the North Atlantic that could explain the variations in the presence of the intermediate water masses in the GoC. Focussing on the greater presence of the AAIW in autumn, it seems that the following
two types of processes are required (Fig. 12):

a) Processes that promote the transport of intermediate waters (AAIW and SAIW) from high latitudes and low latitudes toward the GoC.

b) Processes that may favour the northward transport of intermediate waters when these have arrived at latitudes close to the GoC.

Regarding the first type of processes, we detach the small scale variability of the wind stress curl close to the continents and its implications in the near continent meridional Sverdrup transport. As shown in Figure 9b, these transports southward along the Portuguese coast and northward along the African coast, maintain a permanent direction during the whole year but become more intense from June to September along the African coast and from October to February along the Portuguese coast. Therefore, during summer the conditions for a decisive northward progression of the AAIW along the African coast
are met and this mechanism could be responsible for bringing the AAIW up to the Canary Islands region (Fig. 12). However, since these transport processes diminish considerably in September, they may not be able to sustain the subsequent northward progression of the AAIW that could explain the observed maximum in the percentage of the AAIW along the slope of northern Morocco and the GoC attached in November. Therefore, it is then necessary to find additional processes that help sustain this northward progression up to the GoC. This leads us to the second type of processes.

Regarding the second type of processes, we will consider the seasonal variations of the Ekman pumping in the zone of the subtropical gyre (Fig. 12) closest to, and within the GoC. Figure 9c shows these values for the different climatological months. Note that when the Ekman pumping becomes less negative to the west of the GoC, southward transport on the eastern branch of the gyre weakens and this could favour a greater spreading of the AAIW towards the north. On the one hand, this weakening of the negative Ekman pumping in this zone is more pronounced during March-June and September-

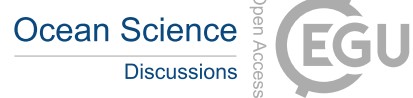

November. On the other hand, the Ekman pumping inside the GoC is positive during the whole year. As previously described, this permanent divergence in the Ekman layer caused by a small scale cyclonic wind pattern existing in this area and more intense during summer, would favour the displacement of the water masses underneath the Ekman layer toward the gulf, exerting a suction effect on them. Therefore, intermediate waters that arrive at a latitude close to the gulf could continue

progressing northward inside the gulf thanks to this effect. Note that this Ekman pumping experiences a significant increase in November after having reduced its intensity during September and October.

### 4.2 Implication of the AAIW on the morphology of the continental margin

Intermediate water masses have been shaping the morphology along the northern Moroccan coast, the GoC and the Atlantic Iberian continental margins (see compilations in Hernández Molina et al., 2011, 2016). During the last decades numerous

researchers and institutions have analysed the influence of both the MOW and ENACW on the sea floor morphology with great detail, particularly within the GoC (e.g., Madelain, 1970; Melières 1974; Nelson et al., 1993, 1999; Hernández-Molina et al., 2003; 2011, 2014 and 2016; Llave et al., 2007; Garcia et al., 2009; Roque et al., 2012, Sánchez-Leal et al., 2017, among many others). These authors identified large depositional and erosional (contourite) features along the middle and upper continental slopes due to the vertical and lateral variation of the aforementioned intermediate water masses. The

results obtained in the present study reveal clear seasonal variations of the AAIW and its interrelation with the SAIW, MOW and ENACW. Nevertheless, changes in distribution of these intermediate water masses and the implications in particular of the AAIW for shaping the morphology and controlling sedimentation along the northern Moroccan coast, Gulf of Cádiz and the Atlantic Iberian margins has not been considered so far and two new questions may arise: Is there any influence of the AAIW along the continental slope at present? Was there any influence of these intermediate water masses in the past?

Some recent papers have described the occurrence of bottom current features along the middle slope of the northern Moroccan margin, which do not fit well with present circulation of the ENACW and the MOW and its associated interphases (pycnoclines). The onset of these features has been related both to the action of deep tides or a branch of upper MOW veering southwards off the Strait of Gibraltar along the Moroccan margin (e.g., Vandorpe et al., 2014, 2016; Lebreiro et al., 2018). Deep (barotropic and baroclinic) tides are important secondary processes amplifying the background water mass

circulation generating local and smaller features but are not relevant for large regional depositional or erosional feature formation along continental slopes (Rebesco et al., 2014; Hernández-Molina et al., 2016). In contrast, a branch of upper MOW veering southwards off the Strait of Gibraltar along the Moroccan margin is apparently against the Coriolis forces in the area (Baringer and Price, 1997, 1999) as well as the reported MOW regional circulation (Sánchez-Leal et al, 2018). Northward flowing of a vigorous AAIW close to the margin, as reported here, could be considered as a plausible regional

control factor in developing those features.  However, this would require more detailed sedimentary, morphologic and paleoceanographic work in order to confirm this hypothesis.

Vertical variations of the MOW and ENACW have been reported at short (seasonally) scales (e.g., Borenäs et al., 2002, Bellanco and Sánchez-Leal, 2016) and in longer geological cycles (Llave et al., 2006; Voelker et al., 2006; Rogerson et al.,



2012; Hernández-Molina et al., 2016; Lofi et al., 2016 among other). The Pliocene and Quaternary (last 5.3 Ma) sedimentary evolution of the slope has been related to those vertical changes. For example, in glacial periods some authors have reported that the MOW was flowing approximately 700 m deeper than today with associated changes in both East North Atlantic Deep Water (ENADW) and North Atlantic Deep Water (NADW) (Schönfeld and Zahn, 2000; Schönfeld et al., 2003,

Rogerson et al., 2012). An open debate about the relationship between the variation in the paths of these water masses and the sedimentary evolution along the northern Moroccan coast, GoC and the Atlantic Iberian continental margins has been ongoing since the recent Integrated Ocean Drilling Program (IODP) Expedition 339 in the GoC (Expedition 339 Scientists, 2012; Stow et al., 2013b; Hernández-Molina et al., 2013; 2016). However, in this debate the influence of the AAIW in the past has not been taken into account.

The AAIW represents a cold intermediate water mass formed at the ocean surface in the Antarctic Convergence Zone/Antarctic Polar Front (between 50° to 60°S), mainly at the southwest of the southern tip of South America (Tomczak and Godfrey, 2003). After its formation the AAIW flows north (Fig. 12) as an intermediate water mass as far as 20°N, with trace amounts as far as 60°N. It continues northward until it encounters other intermediate water masses (Talley, 1999). This is the case for the AAIW in the Atlantic Ocean (and similar in the Indian Ocean) where this water mass has higher influence

and is denser in comparison with the Pacific Ocean where the AAIW is comparatively less important (Bostock et al., 2013). The formation and circulation of the AAIW is an important component of the upper branch of the AMOC that is associated with the transport of heat and salt within the Southern Hemisphere subtropical gyre (Stramma and England, 1999). Different authors have reported enhanced AAIW circulation during colder periods at different scales (e.g., Oppo and Horowitz, 2000; Curry and Oppo, 2005; Pahnke et al., 2008; Muratli et al., 2009; Makou et al., 2010; Jung et al., 2011; Wainer et al., 2012

among other). For example, during the last glacial, there is increased formation of intermediate water and the production of NADW is significantly weakened, and the NADW is replaced in large extent by enhanced AAIW (Wainer et al., 2012), which circulated lightly with a main core ~1100 m in a depth range between 900 and 1,270 m (Makou et al., 2010). Therefore, vertical and lateral variations of the AAIW during glacial vs interglacial periods have been reported (e.g.: Viana et al., 2002; Bozzano et al., 2011; Preu et al., 2012; 2013; Voigt et al., 2013), and some authors demonstrated that during the

last decades the AAIW has been reduced, significantly warm (0.058–0.158°C / decade) and shoaling (30–50 dbar / decade) since becoming less dense [up to 20.03 (kg/ m3) / decade], due to global warming (e.g., Downes et al., 2010; Shmidtko and Johnson, 2012).

Therefore, based on all the above considerations we can conclude that the paths of the AAIW had to vary throughout the geological past at different scales, increase during cold periods and decrease and shoal during warmer periods. The results

presented in this paper are important and any future discussion on this subject should consider the lateral and vertical interaction of intermediate water masses in general and evaluate the role of AAIW in the past in particular, since its influence could be important, especially related to colder periods, in controlling the sedimentation along the slope.  For example, in contributing to higher siliceous production in determinate time periods as has been described with Expedition 339 in the GoC (Expedition 339 Scientists, 2012; Stow et al., 2013b; Hernández-Molina et al., 2013; 2016).



## 5. Conclusions

The analysis of the seasonal variation of the intermediate water masses carried out in the Gulf of Cádiz and adjacent areas has determined remarkable changes of the Antarctic Intermediate Water (AAIW) and the Subarctic Intermediate Water (SAIW). During autumn a greater presence of the AAIW, coming from lower latitudes, seems to be related to a reduction in
the presence of the SAIW and the ENACW. This interaction also affects the Mediterranean Outflow Water(MOW) which is pushed by the AAIW toward the upper continental slope. In the rest of the seasons, the SAIW is the predominant water mass reducing the presence of AAIW.

This seasonal variability in the interchange between these intermediate water masses can be explained based on the concatenation of several wind-driven processes acting during the different seasons. The summer intensification of the
Sverdrup transports near the African coast makes the progression of AAIW possible from low latitudes up to latitudes above the Canary Islands. Subsequently, once the AAIW has reached latitudes close to the gulf, its northward transport is sustained thanks to a decrease in the intensity of negative Ekman pumping within the subtropical gyre west of the Gulf of Cádiz, which presents its minimum intensity during autumn (Fig. 12). This transport toward the Gulf of Cádiz is also favoured by the permanent cyclonic wind system that dominates locally in the GoC, although it shows its highest intensity in the months of
July to August. In November it again experiences a significant increase. The high percentage of the SAIW during winter, spring and summer could be explained by the permanent southward Sverdrup transport that occurs near the coast of the Iberian Peninsula which favours the arrival of the SAIW to the gulf (Fig. 12). From September onward this transport begins to weaken due to the arrival of the AAIW from the south and the deviation of part of the SAIW transport toward the eastern side of the subtropical gyre (west of Gulf of Cádiz), where the intensity of the negative Ekman pumping is reduced between
September and November.

Our results are important for a better understanding of intermediate water mass variability along the northern Moroccan coast, Gulf of Cádiz and the Atlantic Iberian margins but further paleoceanographic, sedimentary and morphological research is needed in order to decode changes in the geological past and determine how these water masses, in particular the AAIW, have been shaping the morphology and controlling the sedimentation along the northern Moroccan coast, the Gulf of
Cádiz and the Atlantic Iberian margins.



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



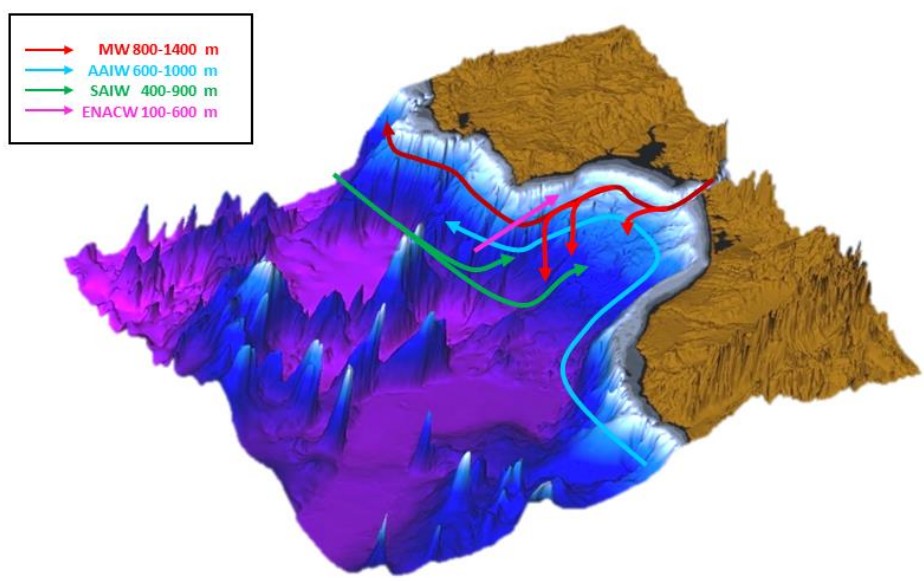

**Figure 1. Illustration of intermediate water mass circulation in the Gulf of Cádiz. The red line indicates the MOW along the Spanish and Portuguese coast. The blue line is for the AAIW coming from the African coast from the south. The green line is for the modified SAIW coming from the northwest of the Iberian Peninsula. The purple line is for the ENACW coming from the western part of the Gulf of Cádiz.**





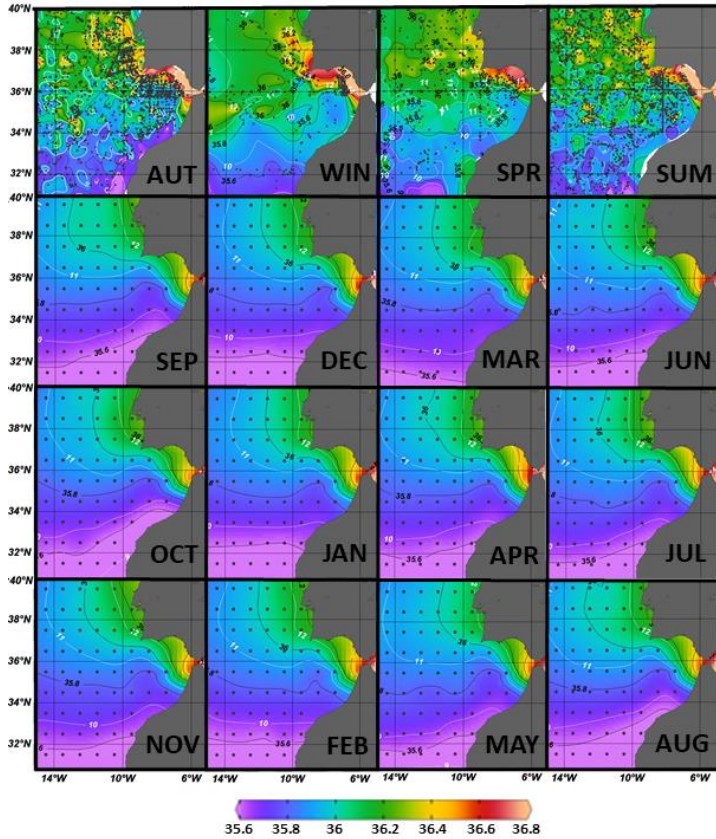

**Figure 2. Observed salinity on the isopycnic surface σ= 27.5 for the different seasons (top row). The rows below show the same distributions for each climatological month as shown by WOA data. The white lines represent the temperature values a density σ= 27.5.**





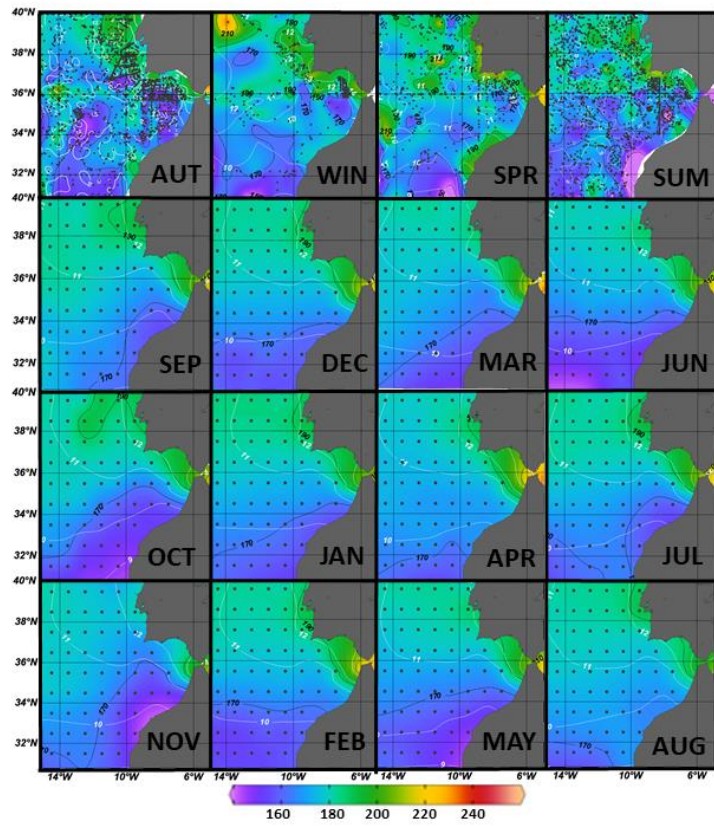

**Figure 3. Observed oxygen concentration on the isopycnic surface σ= 27.5 for the different seasons (top row). The rows below show the same distributions for each climatological month as shown by WOA data. The white lines represent the temperature values a density σ= 27.5.**



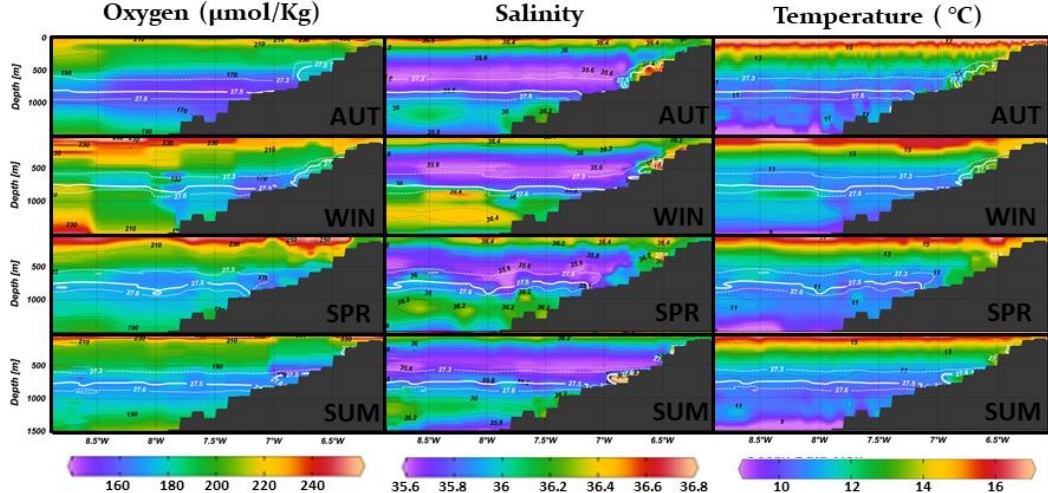

**Figure 4. Vertical sections disposed zonally at a latitude of 36 ° N of observed variables for each season. From left to right: Oxygen concentration, Salinity and Temperature. From top to bottom: the different seasons.**



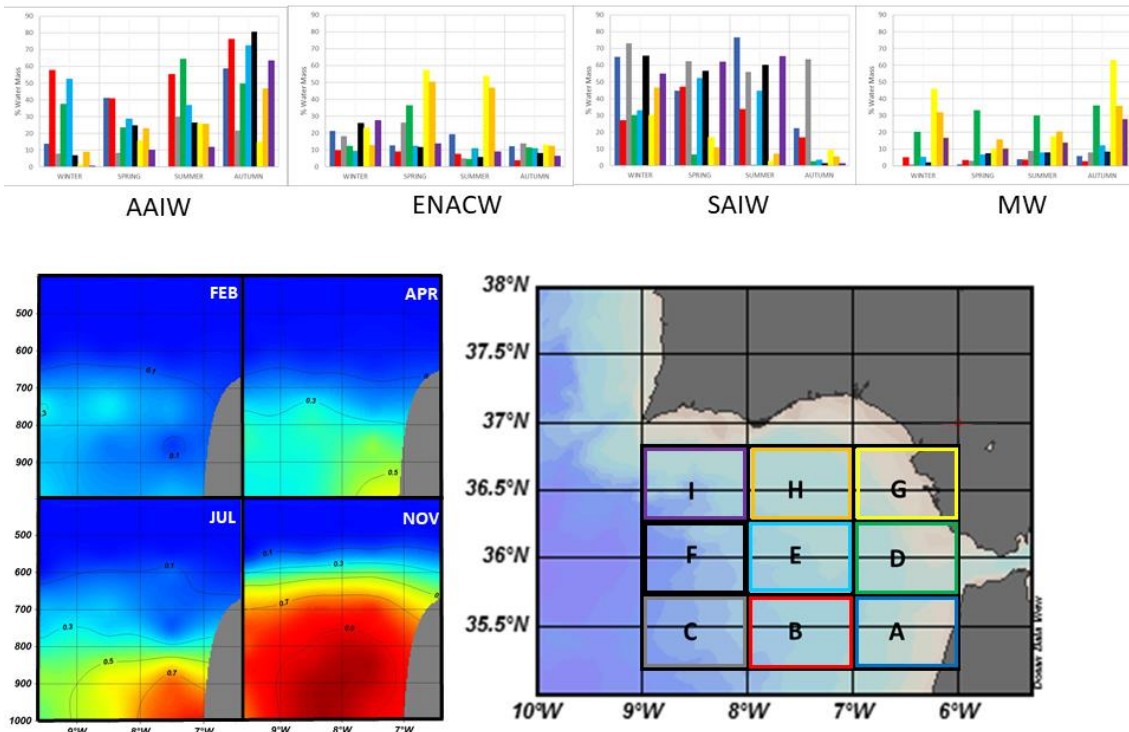

**Figure 5. Values of percentages of each water mass resulting from the OMP analysis in each of the boxes shown in the map in the right hand lower corner, for each season. In the left hand lower corner the resulting percentages are shown from the OMP analysis applied on the WOA data base for a zonal section at 36o N in selected climatological months. The data selected for OMP analysis are those in the depth range of 400-1000 m.**



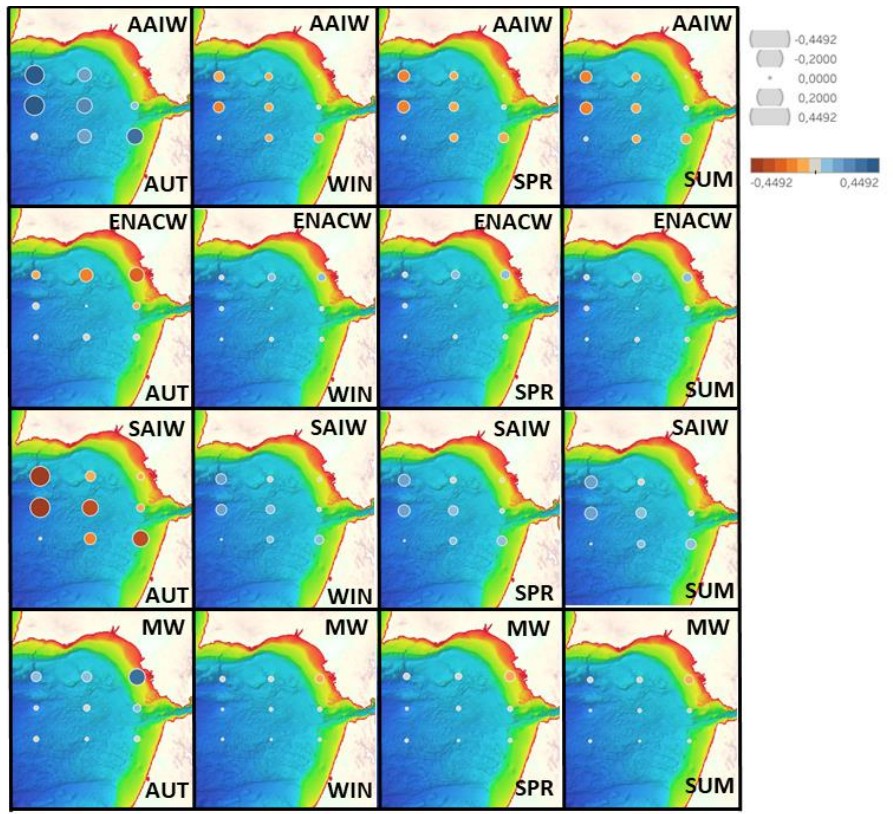

**Figure 6. Seasonal variation of the percentages of each water mass given by the first principal component resulting from the PCA on the percentages previously estimated in the nine boxes considered in the GoC.**




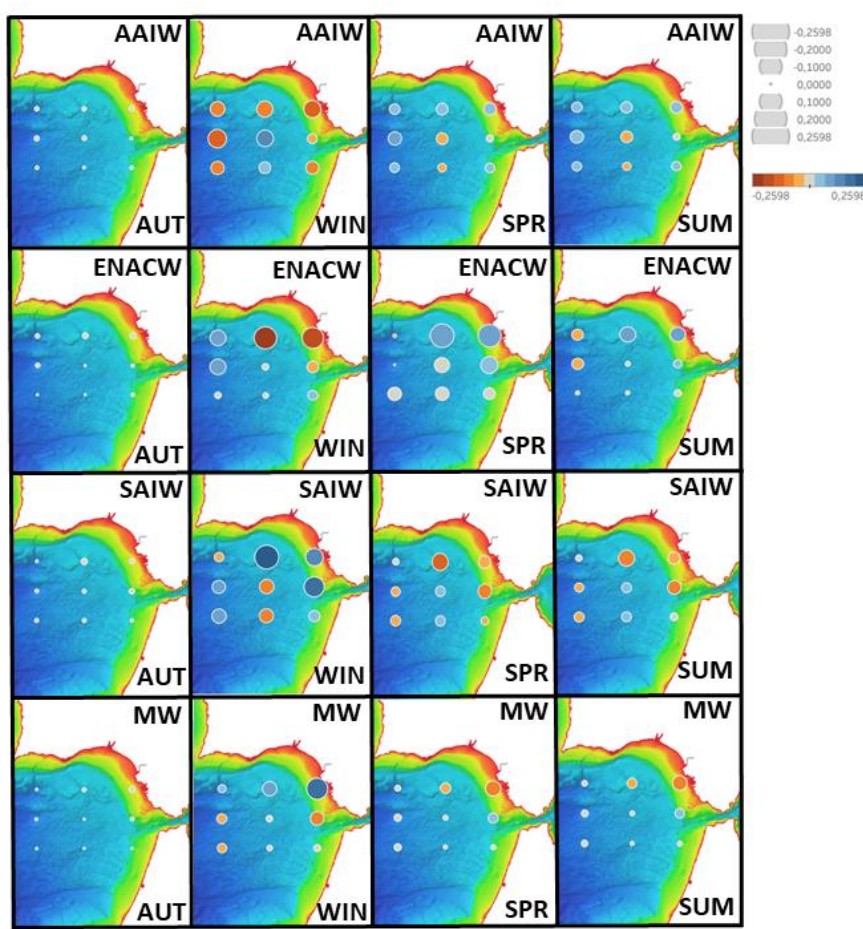

**Figure 7. As figure 6 but for the second principal component.**



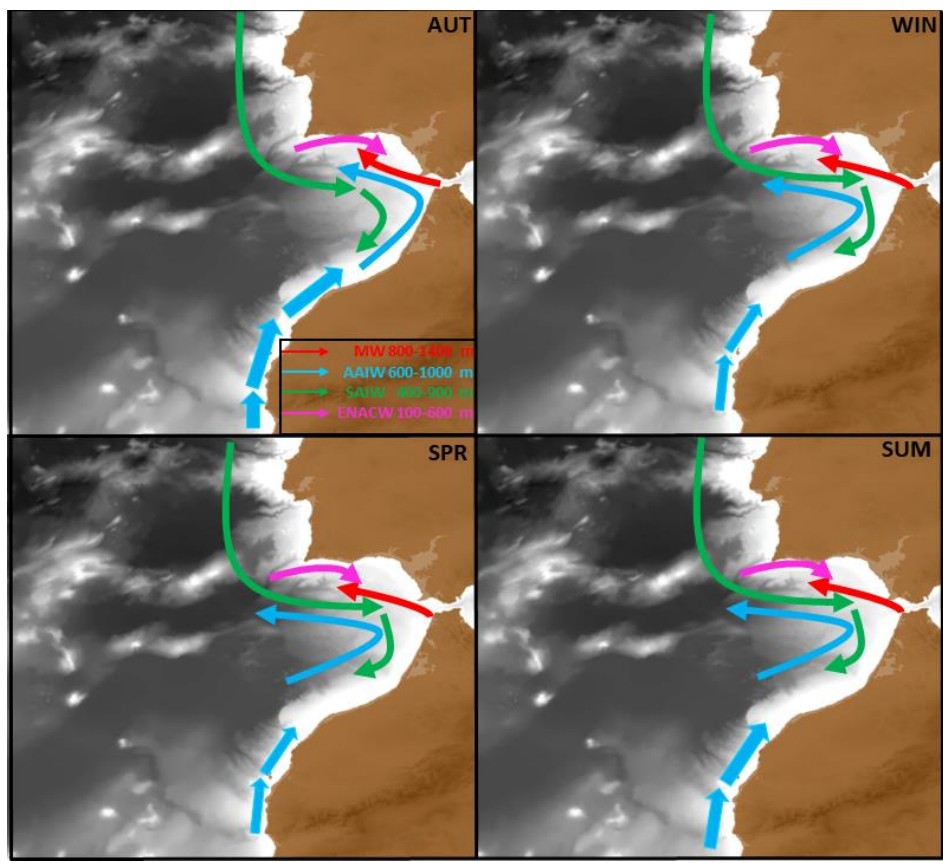

**Figure 8. Illustration of the preferential tracks of the different intermediate water masses throughout the different seasons.**





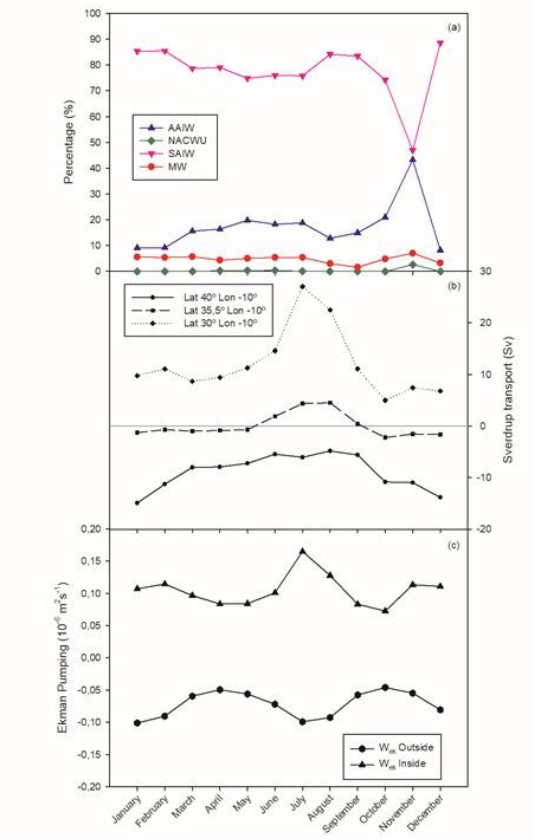

**Figure 9. a) Averaged percentages of each water mass resulting from the application of OMP analysis on WOA data, for each climatological month, at 35.5 °N in the Gulf of Cádiz. b) Sverdrup transport at three locations close to the continental borders: south of GoC (30 °N, -10 °W), in the GoC (35.5 °N, -10 °W), north of GoC (40 °N, -10 °W). c) Monthly Ekman pumping in two zones; Outside the GoC (between 35°-36° N and 19°-20° W) and Inside the Gulf of Cádiz (between 35°-36° N and 7°- 8° W).**



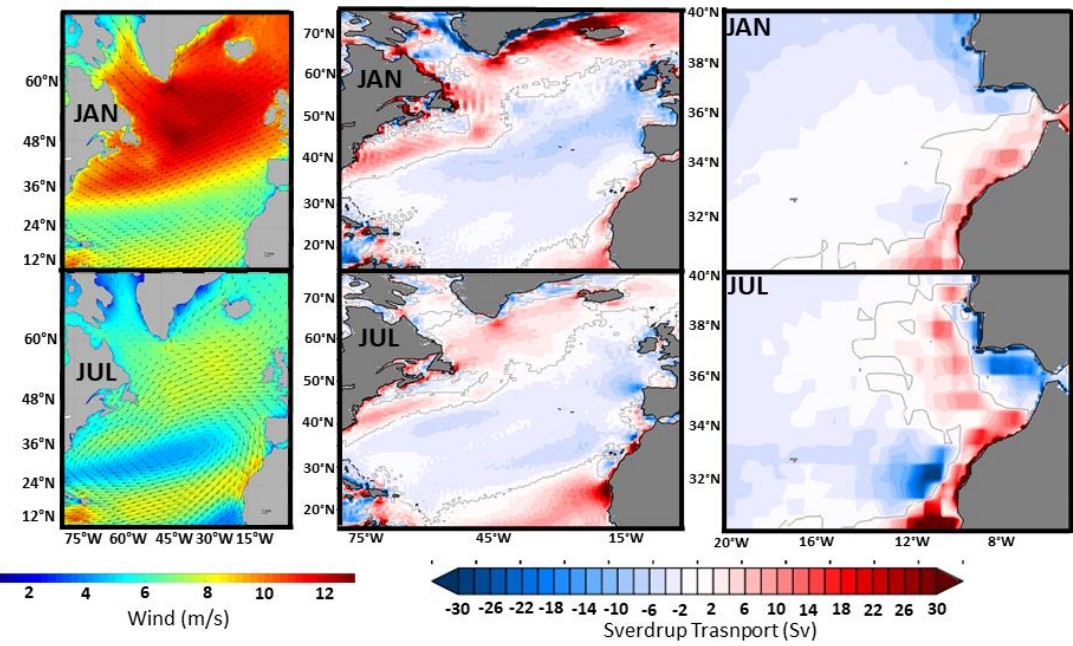

**Figure 10. Maps of surface winds and Sverdrup transport in January (upper) and July (lower).**

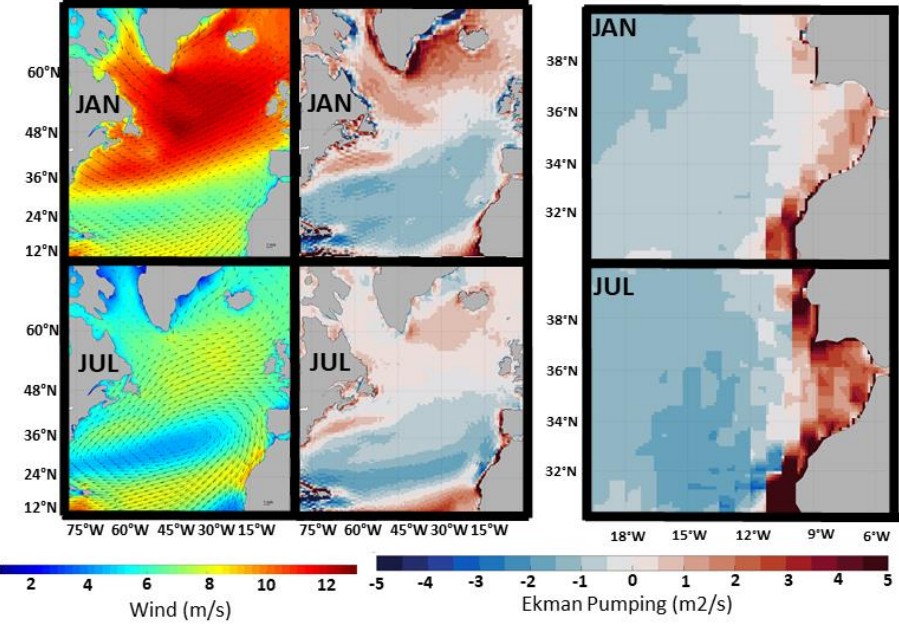

5   **Figure 11. As figure 10 but for Ekman pumping.**





**Figure 12. Sketch of the proposed mechanism for explaining the seasonal variability of the intermediate waters along Gulf of Cádiz (GoC) and surrounding areas. The position of the main gyres in the Atlantic, the Intertropical Confluence Zone (ITCZ) and circulation of the AAIW, SAIW, ENCW and MOW are indicated. (Compilation based on; Krauss et al., 1987; Stramma and Siedler, 1988; Faugères et al., 1993; Ior and Lozier 1999; Pickart et al., 1999; Stramma and England, 1999; Coulbourne and Foote, 2000; Lavander et al., 2000; Rhein, 2000; Fratantoni, 2001; Raymo et al., 2004; Mackie, 2005; Plez et al., 2005, 2009; Rahmstorf, 2006; Stein, 2007; Gil et al., 2008; Yashayaev, and Clarke, 2008; Hernández-Molina et al., 2011 and references therein; Cheng et al., 2012; Fiz et al., 2013)**