# Peer review of "Seasonal variability of intermediate water masses in the Gulf of Cadiz: implications of the Antarctic and Subarctic seesaw."

_Ocean Science, 2019_

## Referee Comment (RC1) · Anonymous Referee #1 · 12 May 2019

General comments The main goal of this paper is to explain and relate the seasonal variations of the presence of Antarctic and Subarctic intermediate water masses in the Gulf of Cadiz and adjacent Atlantic region and their implications on the morphology and sedimentation along the local continental slopes. The gathering of a large set of observational data in the region complemented by climatological fields from the World Oceanographic Atlas allowed an analysis of the seasonal variability of intermediate water masses relevant in the region. As the authors are convinced that the results might contribute to a better understanding of the role, in the geological past, played by those intermediate water masses, the present results are worth to be published, after some corrections. In general, the text is clearly written and most of the figures

complement it.

Specific comments Page 1, Title: "Antarctic and Subarctic seesaw model". The reference to a "seesaw model" in the title but nowhere else in the text should be avoided. Page 1, Abstract: The sentence "the seasonal variability for the predominance of these intermediate water masses is explained by a novel model based on the concatenation of several wind-driven processes acting during the different seasons", overrates somehow the importance of the methodology used in this work. Page 2, lines 28, 29: Since MW is characterized by low nutrients, its interaction with AAIW, (characterized by high nutrients) can only cause an increment on the MW nutrients (and not the other way around as expressed in the manuscript). In what concerns the dissolved oxygen, as both AAIW and MW have low concentrations, it is not clear how can their interaction increment the AAIW concentration. Page 7, line 1: in fact, the AAIW penetration in the GoC seems "more accentuated and closer to the African coast" in autumn if we look at the salinity distribution; however, from the oxygen distribution this seems to happen in summer. Page 7, line 3: It is not clear that the monthly distributions shown by the WOA data always agree with the observed data behaviour Page 7, line 7: Although the oxygen concentration shows the major entrance of the AAIW close to the African coast occurring in November, this is not clear from the salinity distributions. Page 7, line 8: "...restricted during the spring months". In fact, this is shown only in March by the salinity distribution and only in April by the oxygen distribution. Page 7, lines 16, 17: the salinity and temperature sections are not discussed in the text but only the oxygen distribution. Page 7, lines 24-26: The SAIW shows its lowest percentages (less than 10%) in boxes closer to the Iberian coast (from E to I) during autumn, while the highest values are found in winter in these boxes. The ENACW in the closer to coast boxes (G, H) shows its highest values in spring and summer... Page 9, lines 24, 25: looking at Fig. 11, the referred latitude values, 50° and 30°N, could perhaps be substituted by 48° and 36°? Page 11: this is the only page in the text where NAO is referred as a cause for the N Atlantic gyre' displacements. The rest of the text mentions only seasonal variations of the gyres position. Attention should be made to the fact that NAO is not a

seasonal process and so NAO and seasonal effects should not be confused. Page 32: Figure 12 seems to be considered by the authors a good illustration of the processes referred in the text. It shows the seasonal (summer versus winter) displacement of the North Atlantic gyres and the paths of water masses (not all relevant in the text) but there are other depicted processes, like "upwellings", ITCZ displacement (July versus January), which are not even mentioned in the text. I wonder how relevant and helpful this figure is in the present context.

Technical corrections Page 1 and following: in the general oceanography literature, the water mass found in the North Atlantic and originating in the Mediterranean basin is called Mediterranean Water (MW) and not Mediterranean Outflow Water (MOW). In the present manuscript, the acronym MOW is used in the whole text but is MW that appears in the figures (1, 5, 6, 7, 8, and 9). Page 2, line 16: Michael and McCartney 1992 Page 2, line 17: Brogueira (use only the surname) and Gonçalves 2002 Page 2, lines 19, 29: van Aken (use only the surname) Page 2, line 23: define what is meant by "upper MOW core" Page 2, line 24: Bellanco and Sánchez-Leal (use only the surnames) Page 2, line 25: westward (?) branch Page 3, lines 4, 5, 6: (use only the surnames) Page 4, line 15: linear equations Page 6, line 19: best characterizes Page 7, line 5: those maps Page 7, line 14: "was already reported (Tsuchiya...)" Page 7, line 23: Only boxes G (the closest...) and C (the farthest...) show always AAIW percentages... Page 7, line 27: On one hand... Page 7, line 31: zonal section at a latitude... Page 8, line 1: water mass has been Page 8, line 4: percentages of AAIW for... Page 8, line 21: ENACW Page 8, line 23: interpretation of these results Page 9, lines 5 and 6: ïĄšin the symbols for density and Coriolis parameter, the index o should be written in lower case Page 9, line 8: Figures 10 and 11 show, respectively, . . . and the Ekman pumping. . . Page 9, line 16: 35.5°N Page 9, line 23: . . . the zone where convergence (negative Ekman pumping) is more important within the subtropical gyre. . . Page 9, line 27: a displaced towards the south gyre. . . Page 9, line 28: divergence (positive Ekman pumping) . . . Page 10, line 8: decrease significantly Page 10, line 31: resolution of 1° Page 11, lines 1, 2: Velez-Belchi et al., 2017 Page 11, line 5: why referring to Fig. 12 when "NAO and

other atmospheric regimes" are mentioned in the text? In Fig. 12 only summer and winter positions of the gyres are represented. Page 11, line 16: from high latitudes and low latitudes? Both AAIW and SAIW come from high latitudes. Page 11, line30, 31: Ekman pumping within the GoC and in the zone of the subtropical gyre closest to GoC Page 12, line 18: margins have not... Page 13, lines 8 and 34: Stow et al., 2013 (not 2013b) Page 13, line 19: Muratli et al. 2009? or 2010 (as in the References)?; Jung et al., 2011? or 2010 (as in the References)? Page 13, line 20: among others Page 13, line 22: 1270 m Page 13, line 25: significantly warmer Page 14, line 13: why reference to Fig. 12 for the minimum intensity in autumn? Page 18, line 1: sédimentaire Page 18, line 5: undercurrent sandy Page 19, line 17: quantitative assessment Page 21, line 14: Clarke

References: Papers referred in the text or in the figure' legends but not present in the References: Ambar and Howe, 1979 Hernández-Molina et al., 2003, 2011, 2016 Lebreiro et al., 2018 Sánchez-Leal et al., 2018 Stramma and England, 1999 Vandorpe et al., 2014, 2016

Papers present in the References but not referred in the text: Pérez et al., 2013 Schlitzer, 2018

Figures and legends: Figs. 2 and 3 legends: at density 27.5 Fig. 4: ïĄ■mol/kg (and not ïĄ■mol/Kg) Fig. 5 legend: 36°N; I would suggest to swop places between the map and the percentage distributions. The text along the axes of the graphics with the percentages should be clearly seen. Fig. 6 and 7: explain the meaning of the numbers and figures just above the colour scale Fig. 8: why the arrows for the AAIW along the African coast are wider in autumn and summer? Fig. 9: why -10°W instead of 10°W? ; "in two zones: ..." Fig. 10 legend: mention the zoomed region; in the scale, Sverdrup Transport (and not Trasnport) Fig. 11: the graphics should have the same dimension as in Fig. 10; units of Ekman pumping should be m/s. Fig. 12 legend: Iorga and Lozier; Peliz et al., 2005; Pérez et al, 2013 (and not Fiz et al.); ENACW (instead of ENCW)

---

## Referee Comment (RC2) · Anonymous Referee #2 · 21 Jul 2019

Dear Prof. Stevens,

Roque et al. present an interesting study on the influence of Antarctic Intermediate Water (AAIW) on the seasonal hydrographic changes within the Gulf of Cadiz. These findings certainly present an advancement on the current understanding of the influence of AAIW and its interactions with Mediterranean Outflow Water (MOW); the latter is considered an important modulator of North Atlantic hydrography and its meridional overturning circulation. As a paleoceanographer myself, these findings could be potential extremely interesting for studies on the MOW behavior throughout the geological past and reach a wide readership. Although the manuscript is overall well written, I do

have some questions I would like to invite the authors to address in a revised version of this manuscript:

In the beginning, the authors define four major water masses that are the basis for their analysis and discussion. However, I am bit confused by their terminology. In past studies, the Eastern North Atlantic Central Water (ENACW) was separated in a subtropical and a subpolar fraction (i.e. Voelker et al., 2015). Yet, the authors only define ENACW without specifying its origin. The authors instead define Subarctic Intermediate water (SAIW) which I am not sure what the difference is between the previously reported ENACW of subpolar origin. Perhaps the authors could clarify this for the reader. I think this would help to relate their study better to previously published papers on the modern hydrography of the region.

Secondly, the authors argue that AAIW "pushes" MOW up the shelf during autumn. I was wondering how these findings relate to the seasonal changes in MOW density. During autumn, MOW reaches its annual density minimum (Millot et al. 2006). Though my question is, does the MOW flow higher up on the shelf during autumn due to the AAIW pushing it up or does the less dense MOW simply settle higher up on the shelf by itself, and thus allows AAIW to extend vertically within the water column?

Generally, I feel the authors missing the opportunity here to also analysis their data sets for any decadal patterns. For instance, I would be very interested in knowing if the authors can make any statement regarding the temporal stability of the relationship between AAIW intrusion into the Gulf of Cadiz and its interactions with MOW. Millot et al. (2006) argued that MOW become more saltier and warmer after the 1990s. The change in MOW characteristics between 1960-1980 vs the 1990-2000s is often used as an analog for glacial-interglacial changes of MOW conditions. Hence, I am wondering if the authors see any change in AAIW presence and simultaneously MOW settling depth between 1950-1980 and 1990-2000s? I feel that including this kind of temporal information could substantially enhance the current discussion under section 4.2. where the authors try to hint at this relationship and its possible interest for paleoceanographers but do not provide any real new insights.

Minor comments

Please use, if not otherwise requested by OS, continuous line numbering across all pages for the next version. I find the current page wise numbering a bit confusing.

I feel the introduction might need a bit more rephrasing as some of it reads a bit confusing and repetitive at times.

There are a number of typographic issues throughout the manuscript as well as some issues with the format of the citations. I picked the ones out I could find but please check this carefully again.

p.1 Line 13: 4 is italic when the others are not.

p.2 Lines 2 to 4: Please rephrase this sentence. It is very long and contains to many information. Please break it up into at least two sentences. Although is meridional heat transport not the same as MOC? Does meridional freshwater transport not imply THC?

p.2 Lines 4: please subscript the 2 in CO2.

p.2 Line 4: What do you mean by "their"? THC or MOC or any other transport you listed previously? Specify.

p.2 Line 9: Citation missing for the depth informations

p.2. Line 12-13: What do you mean by "cold" and "warm" periods? Interglacial and glacial? Stadial and interstadials? Specify.

P2. Line 14 to 15: The sentence starting with "The AAIW is well..." feels completely out of place here. In the sentence above you were stating something about warm/cold periods and now we suddenly jumped to seasonality? Please rephrase this sentence!

p.2 Line 16: Why is the GOC not included in the brackets with the Citation?

p.2. Line 22 to23: What is the difference between MOW and upper MOW suddenly? I
do not understand why the upper MOW is singled out here right now.

p.2. Line 24: Is this the right citation style? First names are usually not included in citations! Please check this with the OPS guidelines.

p.2 Line 25 to 26: Is this not more or less the same information as provided in Lines 15-17. Please check if this sentence and the above-mentioned sentences one cannot be spliced together.

p.2 Line 33: Please delete the comma between regions and progresses.

p.3 Line 3: double space in front of the Citation.

p.2 Line 4 to 5: Again, check the Citation style! See comment above!

p.3 Line 14: Please replace "near" with "vicinity".

p.3 Line 19: Is the "the" really necessary here? Maybe just "Data set".

p.3 Line 21: please add a "the" between "and" and "surrounding".

p.3 Line 21: what do you mean by surrounding areas? Specify.

p.3 Line 23: how many cruises? In what years did they take place? and what do you mean by "mostly carried out by the University of Cadiz"? If they were not carried out by this University then list the institution that carried them out. Please provide enough information for the reader!

p.4 Line 3: It might be useful to add the expeditions also in Table 1. The Table caption should be listed above the table unless OS guidelines suggest otherwise.

p.4. Lines 5 to 9: Did you compare the wind data between 1979-2018 only to hydrographic data for the same time frame or did you use the entire data set from 1900 to 2013 for that? If you used the entire time series of the hydrographic data set would that not be problematic given the effects that global warming had on wind field changes in the last decades?

p.4 Line 10: I think it should be "the" and not "an" Optimum Multi. . ..

p.4 Line 11: What do you mean by "each data set"? Each data set separated by data base origin? Yearly data across all data bases? Seasonal data? Please specify.

p.4 Line 15: ad an "s" to equation.

p.5 Line 9: Caption of Table 2 should be above the table I think.

p.5 Line 11ff: Generally, I am missing some statements here regarding the data pre-processing steps, and the software (e.g. MATLAB?) used for the analysis. To the best of my knowledge PCA requires the data to be gaussian distributed prior to analysis. How did you pre-process your data?

p.6: Line 18 to 24: This is does not like a result. Does this paragraph really require a section header and number? I find this bit somewhat confusing.

p.7 Line 14: double space before Citation

p.7 Line 20: What do you mean by ". . .seems to have a certain intra-annual variability,..". Please elaborate on this if you consider it important; if not rephrase it.

p.8 Line 11: Should it not read "PCA" instead of "PC"?

p.8 Line 12ff: Just for clarification for me the percentage of explained variance of the PC′s is the number stated in Figures 6 and 7?

p.8 Line 30: double spacing before Citation.

p.9 Line 10: In the beginning you defined GoC as Gulf of Cadiz but you jumping back and forth throughout the entire manuscript between using the abbreviation or not. Please streamline this.

p.10 Line 16: double spacing before Citation.

p.10 Line 17: I would change "Until now" to "Thus far".

p.10 Line 31: I think this is a typo by resolution of "1o" of the wind. . . do you mean 10?

p.12 Line 4: substitute "gulf" with GoC or Gulf of Cadiz.

p.12 Line 12: the ";" between 2003 and 2011 should be a ",".

p.14 Line 12: replace "thanks" with "due".

p.14 Line 19: include "the" between "west" and "Gulf of Cadiz".

---

## Author Comment (AC1) · 25 Aug 2019

Authors replies to the interactive comments of anonymous referee #1 (12 May 2019) on "Seasonal variability of intermediate water masses in the Gulf of Cadiz: implications of the Antarctic and Subarctic seesaw model" by David Roque et al.

**RC:** denotes referee's comments

**AR:** denotes authors' reply

**RC:** Page 1, Title: "Antarctic and Subarctic seesaw model". The reference to a "seesaw model" in the title but nowhere else in the text should be avoided.

**AR:** Reviewer is right, we have remove the term model from the title.

**RC:** Page 1, Abstract: The sentence "the seasonal variability for the predominance of these intermediate water masses is explained by a novel model based on the concatenation of several wind-driven processes acting during the different seasons", overrates some- how the importance of the methodology used in this work.

**AR:** Reviewer is right, we have redrafted these line of the abstract expressing the subject in a more objective fashion.

**RC:** Page 2, lines 28, 29: Since MW is characterized by low nutrients, its interaction with AAIW, (characterized by high nutrients) can only cause an increment on the MW nutrients (and not the other way around as expressed in the manuscript). In what concerns the dissolved oxygen, as both AAIW and MW have low concentrations, it is not clear how can their interaction increment the AAIW concentration.

**AR:** Reviewer is right. We have redrafted this part of the text and have removed the phrase referring to this subject. It was a mistake.

**RC:** Page 7, line 1: in fact, the AAIW penetration in the GoC seems "more accentuated and closer to the African coast" in autumn if we look at the salinity distribution; however, from the oxygen distribution this seems to happen in summer.

**AR:** Reviewer is right, this part of the text has been corrected accordingly to the reviewer observation.

**RC:** Page 7, line 3: It is not clear that the monthly distributions shown by the WOA data always agree with the observed data behaviour. Page 7, line 7: Although the oxygen concentration shows the major entrance of the AAIW close to the African coast occurring in November, this is not clear from the salinity distributions.

**AR:** We have redrafted the text paying more importance to the coincidence as to the oxygen concentration data of WOA data with the observed data. We have added the phrase

"attending to the oxygen concentration distribution, which offers the most marked differences with the other water masses (see table 2)"

**RC:** Page 7, line 8: ". . .restricted during the spring months". In fact, this is shown only in March by the salinity distribution and only in April by the oxygen distribution.

**AR:** We have added the text 'only slightly suggested in March by the salinity distribution and in April by the oxygen distribution.'

**RC:** Page 7, lines 16, 17: the salinity and temperature sections are not discussed in the text but only the oxygen distribution.

**AR:** As we told in the previous question, we give more confidence to the oxygen distribution than to the salinity distribution as to identified the different intermediate water masses. By these reason salinity distributions are not mentioned.

We have added the text 'Note that salinity and temperature distributions are not helpful to discern between AAIW and SAIW (see table 2).'

**RC:** Page 7, lines 24-26: The SAIW shows its lowest percentages (less than 10%) in boxes closer to the Iberian coast (from E to I) during autumn, while the highest values are found in winter in these boxes. The ENACW in the closer to coast boxes (G, H) shows its highest values in spring and summer. . .

**AR:** It has been corrected accordingly to the reviewer suggestions.

**RC:** Page 9, lines 24, 25: looking at Fig. 11, the referred latitude values, 50∘ and 30∘N, could perhaps be substituted by 48∘ and 36∘?

**AR:** Reviewer is right. This item has been corrected in the text.

**RC:** Page 11: this is the only page in the text where NAO is referred as a cause for the N Atlantic gyre' displacements. The rest of the text mentions only seasonal variations of the gyres position. Attention should be made to the fact that NAO is not a seasonal process and so NAO and seasonal effects should not be confused.

**AR:** Reviewer is right. We have added some lines to better understand this reference.

We have added the text 'While in our case we are not dealing with the NAO variability it is worth noting that the effect on the subtropical gyre dynamics of a positive/negative NAO are fairly similar to the effects that a winter/summer wind forcing produce.'

**RC:** Page 32: Figure 12 seems to be considered by the authors a good illustration of the processes referred in the text. It shows the seasonal (summer versus winter) displacement of the North Atlantic gyres and the paths of water masses (not all relevant in the text) but there are other depicted processes, like "upwellings", ITCZ displacement (July versus January), which are not even mentioned in the text. I wonder how relevant and helpful this figure is in the present context.

**AR:** Reviewer is right. Figure 12 is now the figure 1 and it is called in the introduction in order to help the description of the intermediate water circulation in the Atlantic.

**RC:** Page 1 and following: in the general oceanography literature, the water mass found in the North Atlantic and originating in the Mediterranean basin is called Mediterranean Water (MW)

and not Mediterranean Outflow Water (MOW). In the present manuscript, the acronym MOW is used in the whole text but is MW that appears in the figures (1, 5, 6, 7, 8, and 9).

**AR:** We have changed in the whole manuscript the MOW denomination by MW as the reviewer suggests.

**RC:** Page 2, line 23: define what is meant by "upper MOW core"

**AR:** There is a reference (Ambar and Howe 1979) that we have added in the introduction, which describes the Mediterranean Outflow Water and how it is divided into two cores; the upper and lower.

The rest of minor corrections have been corrected following the reviewer suggestions.

---

## Author Comment (AC2) · 25 Aug 2019

Authors replies to the interactive comments of anonymous referee #2 (21 July 2019) on "Seasonal variability of intermediate water masses in the Gulf of Cadiz: implications of the Antarctic and Subarctic seesaw model" by David Roque et al.

**RC:** denotes referee's comments

**AR:** denotes authors' reply

**RC:** In past studies, the Eastern North Atlantic Central Water (ENACW) was separated in a subtropical and a subpolar fraction (i.e. Voelker et al., 2015). Yet, the authors only define ENACW without specifying its origin. The authors instead define Subarctic Intermediate water (SAIW) which I am not sure what the difference is between the previously reported ENACW of subpolar origin. Perhaps the authors could clarify this for the reader. I think this would help to relate their study better to previously published papers on the modern hydrography of the region.

**AR:** Reviewer is right as to the convenience of clarify this subject. There is no general agreement in the denomination of these water masses along the existing literature. We have added some text in order to clarify these denominations, also considering new references to Voelker et al., 2015 and other works.

New added references:

Rios, A.F. Perez,F.F., Fraga,F., Water masses in the upper and middle North- Atlantic Ocean East of the Azores. Deep Sea Research. Part A Oceanogr. Res.Pap.39 (3-4A), 645–658, 1992.

Voelker A.H.L., Colman A., Olack G., Waniek J. J., Hodell D., Oxygen and hydrogen isotope signatures of Northeast Atlantic water masses, Deep Sea Research Part II: Topical Studies in Oceanography, 116: 89-106, 2015.

**RC:** Secondly, the authors argue that AAIW "pushes" MOW up the shelf during autumn. I was wondering how these findings relate to the seasonal changes in MOW density. During autumn, MOW reaches its annual density minimum (Millot et al. 2006). Though my question is, does the MOW flow higher up on the shelf during autumn due to the AAIW pushing it up or does the less dense MOW simply settle higher up on the shelf by itself, and thus allows AAIW to extend vertically within the water column?

**AR:** Reviewer has provided a very interesting point which deserves to be included in the interpretation of our results about the relation between AAIW and MOW. Unfortunately, our observations do not allow to assess what of the two mechanisms may be behind this behavior. However, taking into account the information published in Millot et al. (2006), the possibility for a less dense MOW be settled closer to the slope leaving room for the AAIW penetrates easily into the Gulf of Cadiz may not be discarded. These ideas are added in the modified version of the manuscript.

New added reference:

Millot C., Candela J., Fuda J.L, Tber Y., Large warming and salinification of the Mediterranean outflow due to changes in its composition, Deep Sea Research Part I: Oceanographic Research Papers, 53 (4), 656-666, 2006.

**RC:** AAIW intrusion into the Gulf of Cadiz and its interactions with MOW. Millot et al. (2006) argued that MOW become more saltier and warmer after the 1990s. The change in MOW characteristics between 1960-1980 vs the 1990-2000s is often used as an analog for glacial-interglacial changes of MOW conditions. Hence, I am wondering if the authors see any change in AAIW presence and simultaneously MOW settling depth between 1950-1980 and 1990-2000s? I feel that including this kind of temporal information could substantially enhance the current discussion under section 4.2. where the authors try to hint at this relationship and its possible interest for paleoceanographers but do not provide any real new insights.

**AR:** Certainly, the suggested subject is very interesting and consequently we have tried to deal with it. Unfortunately, the spatial coverture of the observations is not adequate for performing such analysis. Note in the attached figure, where indicated are the locations of the available casts, the lack of data in the spring and summer seasons for the period 1990-2018.

**RC:** p.2 Lines 2 to 4: Please rephrase this sentence. It is very long and contains to many information. Please break it up into at least two sentences. Although is meridional heat transport not the same as MOC? Does meridional freshwater transport not imply THC?

**AR:** Reviewer is right, we have taken into account his comments and have redrafted this lines accordingly.

**RC:** p.2 Line 4: What do you mean by "their"? THC or MOC or any other transport you listed previously? Specify.

**AR:** Reviewer is right, the phrase has been redrafted and now it leaves clear that we refer to the intermediate water masses.

**RC:** p.2. Line 12-13: What do you mean by "cold" and "warm" periods? Interglacial and glacial? Stadial and interstadials? Specify.

**AR:** In the new version this subject is clarified and specified.

**RC:** p.2 Line 25 to 26: Is this not more or less the same information as provided in Lines 15-17. Please check if this sentence and the above-mentioned sentences one cannot be spliced together.

**AR:** We have removed the duplicity of the information as reviewer suggests.

**RC:** p.4. Lines 5 to 9: Did you compare the wind data between 1979-2018 only to hydrographic data for the same time frame or did you use the entire data set from 1900 to 2013 for that? If you used the entire time series of the hydrographic data set would that not be problematic given the effects that global warming had on wind field changes in the last decades?

**AR:** Reviewer is right in making this question. However, in our analysis the wind data we used match with the WOA data in the section located at 36 º N and not with the observations. We have added some lines to leave more clear this subject in the section 2.1, where we describe the data set.

**RC:** p.5 Line 11ff: Generally, I am missing some statements here regarding the data preprocessing steps, and the software (e.g. MATLAB?) used for the analysis. To the best of my

knowledge PCA requires the data to be gaussian distributed prior to analysis. How did you pre-process your data?

**AR:** The analysis has been performed using the MATLAB function PCA. It is now indicated in the manuscript. In the PCA observations are not imposed to follow a Gaussian distribution. What reviewer must mean is that the observations must be referred to their averaged values before the PCA application. In the we have redrafted this section in order to allow for more clarity in the description of the technique and the way it will be applied to the observations new version on the manuscript.

**RC:** p.6: Line 18 to 24: This is does not like a result. Does this paragraph really require a section header and number? I find this bit somewhat confusing.

**AR:** We think that a previous description of the spatial distribution of the variables along the different seasons is convenient and choose to emplace it as a first subsection within the results section, we think that this option is better than dedicate an additional section dealing with the descriptive analysis.

**RC:** p.7 Line 20: What do you mean by ": : :seems to have a certain intra-annual variability,..". Please elaborate on this if you consider it important; if not rephrase it.

**AR:** We have redrafted the sentence accordingly the reviewer suggestion.

**RC:** p.10 Line 31: I think this is a typo by resolution of "1o" of the wind: : : do you mean 10?

**AR:** Effectively, there is a typo mistake, actually it should read as 1º (one degree) when talk about the Machin and Pelegri (2009) analysis. As we described in the methodology section, the wind data providing by the ERA-interim reanalysis have a much better spatial resolution (0.125 x 0.125 degrees)

The rest of minor corrections have been corrected following the reviewer suggestions.